# A Formulation for a New Environmentally Friendly Varnish for Paintings

Irene Pieralli [1,2], Antonella Salvini [2,*], Eva Mariasole Angelin [3,4], Marisa Pamplona [3], Valeria Cocchetti [5], Giovanni Bartolozzi [1] and Marcello Picollo [1,*]

1 "Nello Carrara" Institute of Applied Physics of the National Research Council (IFAC-CNR), Via Madonna del Piano 10, 50019 Sesto Fiorentino, Italy; irene.pieralli@stud.unifi.it (I.P.); g.bartolozzi@ifac.cnr.it (G.B.)
2 Department of Chemistry, University of Florence, Via della Latruccia 3-13, 50019 Sesto Fiorentino, Italy
3 Conservation Science Department, Deutsches Museum, Museumsinsel 1, 80538 Munich, Germany; eva.angelin@tum.de (E.M.A.); m.pamplona@deutsches-museum.de (M.P.)
4 Chair of Conservation-Restoration, Art Technology and Conservation Science, TUM School of Engineering and Design, Technical University of Munich, Oettingenstr. 15, 80538 Munich, Germany
5 Conservator in Private Practise, Via Maggio 40, 50125 Firenze, Italy; valeria.cocchetti@libero.it
* Correspondence: antonella.salvini@unifi.it (A.S.); m.picollo@ifac.cnr.it (M.P.); Tel.: +39-0554573455 (A.S.); +39-0555226360 (M.P.)

**Abstract:** Laropal® A81 is a urea-aldehyde resin used as a varnish for paintings requiring 30%–40% of aromatic solvents to be dissolved. Considering the dangers of aromatic solvents for the environment and health, an attempt was made to replace them with less impactful and low-toxic solvents. The present research investigates a new formulation with aliphatic hydrocarbons and esters (isobutyl isobutyrate, IBIB) as an alternative to traditional aromatics mixture. Traditional and alternative varnish formulations of Laropal® A81 were studied on an inert support and paint samples to test the resin by itself and in interaction with paint film, respectively. This study aims to compare the two different formulations of Laropal® A81 by evaluating their optical, colorimetric, and stability proprieties before and after natural and accelerated ageing. Colorimetry and spectroscopic techniques (transmittance and reflectance UV-VIS-NIR, FORS, transmittance and total reflectance FT-IR, and $^1$H-NMR) were used for the assessment. Very promising results have been obtained with the application of the alternative formulation, with data comparable to those of the traditional formulation, paving the way for the replacement of the aromatic-based solvents traditionally used with IBIB. This allows a safer and more sustainable conservation practice with considerable benefits for the health of the operators and the environment.

**Keywords:** varnish formulation; Laropal® A81; sustainable solvent; varnish for paintings; spectroscopy; colorimetry

## 1. Introduction

Until the late 19th century, most traditional Western paintings had a transparent varnish coating as a surface finish. This outermost layer had a limited protective role, and its function was mainly aesthetic. In fact, the varnish improved the optical characteristics of the pictorial surface, making it more glossy and smooth, increasing the saturation of the colours [1–5]. An ideal varnish should remain colourless, transparent, removable over time, and should have the correct mechanical properties of strength and flexibility to follow any movement of the underlying substrate without cracking. In addition, the varnish composition should be known and carefully evaluated towards its proper application as it should not alter the integrity of the paint film [1,4,6]. Varnishes for paintings are generally organic-solvent-based, and the formulations are made by solubilising the resin in a low polarity solvent mixture with the highest dispersion factor ($f_d$), related to the percentage of

dispersion forces as defined by the Teas solubility concept, to reduce any interactions with the binder or other pictorial materials [7–13].

Varnish is the layer most exposed to degradation by external agents. Materials originally used as painting varnishes (typically natural resins) can alter at the molecular level mainly due to photo-oxidative degradation, resulting in loss of transparency, yellowing, and change of solubility. For this reason, varnishes are replaced over time during art restoration work, and over the years research has sought to find more stable products [1,4,6]. In the 1920s, synthetic polymer resins, generally considered more stable than natural resins, were introduced to the market. However, varnishes obtained with synthetic polymers often give paints a different appearance from those produced with natural resins [6,9,14]. Starting from the 1980s, studies conducted primarily by de la Rie to identify potential substitutes for natural resins [8,14–16] have shown that low molecular weight (LMW) synthetic resins can produce films with an appearance similar to those with natural resins and have better stability. Nowadays, LMW resins of the hydrogenated hydrocarbon type (e.g., Regalrez® 1094) and the urea-aldehyde type (e.g., Laropal® A81) are widely used in art restoration. These are, respectively, soluble in mixtures without or with a limited content of aromatic solvents [1,6,8,17]. Considering the adverse effects on paint films attributable to aromatic solvents (limiting swelling, solubilisation, and leaching) and the growing concern about the toxicity-related impacts on the environment and humans, the ability to use low polarity solvents potentially less damaging to conservators and paintings is a significant feature [1,8]. Solvent-based varnishes are not particularly resistant to physical degradation, but LMWs, such as Regalrez® 1094 and Laropal® A81, generally retain their characteristics even after ageing. Showing better behaviour than some natural and synthetic polymeric resins susceptible to cross linking processes and thus remaining soluble and easily removable, they are widely used as varnishes for paintings [1,6].

Laropal® A81 was selected for the present study because it is considered one of the most stable resins currently used as a varnish for paintings [16,18]. Laropal® A81 varnish is applied in liquid form by dissolving solid pellets in a suitable solvent mixture (with a certain degree of polarity), and the formation of the film occurs by evaporation of the solvent [15,19–23]. This resin typically forms transparent, colourless, and very stable varnish layers [2,6,24,25] when a solvent mixture with a high percentage of aromatic components is used. Laropal® A81 typically requires 30–40% aromatic content in a hydrocarbon mixture for its solubilisation, whereas a solvent mixture completely devoid of aromatics would not give good results as varnish according to the recommendations [1,2,6]. Manufacturers generally recommend these solvents to improve aesthetic aspects and protective efficacy. However, this indication leads to a critical issue regarding toxicity and sustainability. Indeed, it is well known that aromatic solvents are dangerous both for the environment, due to the processes associated with their disposal as hazardous air pollutants (HAPs), and for the health of the operators, as they are harmful and/or potentially carcinogenic. As such, the need for less impactful and hazardous solvents has paved the way for research on the application of alternative solvents or solvent mixtures [17,26]. The choice of the alternative solvent in the present research stems from previous studies by Eastman Chemical Company Corporate Headquarters [27] on the formulation of protective varnish coatings in the industrial field and by Cremonesi [17] on the use of surfactants and chelators for painting cleaning.

Eastman's 2013 study focused on selecting effective substitutes for xylenes or other aromatic hydrocarbon mixtures used in traditional solvent-based varnish formulations. An aromatic hydrocarbon solvent, such as xylene, is widely used in varnish formulation, either as a primary solvent or as resin diluent. Due to its prominence in the paint industry, it is essential to find less-toxic alternatives to obtain solvent mixtures that have similar performance in terms of polarity (and thus solvent power) and evaporation rate to produce a film with the same characteristics [17]. Eastman found three oxygenated solvents effective for substitution: a long-chain ketone, 2-heptanone or methyl n-amyl ketone (MAK); and two esters, n-butyl propionate (nBuOPr) and isobutyl iso-butyrate (IBIB). They are considered solvents with low-risk toxicity and are not hazardous air pollutants. Furthermore, they have

an evaporation rate that can lead to the production of a film with characteristics comparable to those obtained from xylenes. It has also been shown that the oxygenated solvent can be mixed with an aliphatic hydrocarbon to reduce costs without affecting film appearance or performance [27]. In this research, IBIB was chosen as an alternative solvent among the three proposed by Eastman based on its olfactory properties (the other two have an unpleasant odour) replacing the corresponding percentage of aromatic solvents consisting of a mixture of aromatic hydrocarbons containing xylenes and C9 (e.g., Shellsol® A100, Shellsol® A from Kremer Pigmente) commonly proposed in recipes from the literature [1,2,6].

Cremonesi introduced the use of IBIB in art conservation as part of the research on surfactants and chelating agents for the treatment of paintings [17]. He investigated the possible replacement of xylenes with alternative solvents of intermediate polarity, such as esters (including IBIB), to be added to a mixture of aliphatic hydrocarbon in the solvent phase of cleaning systems. Bearing in mind the issues of carcinogenicity and toxicity of various hydrocarbon solvents, his research considered the use of saturated aliphatic hydrocarbons or mixtures thereof with the lowest possible aromatic content and, in particular, the lowest possible residual benzene content, to be the least dangerous. Cremonesi's goal was to conduct the solubilisation in a safer cleaning procedure for the operator's health.

This research investigated a new varnish formulation for paintings based on Laropal® A81 using mixtures of aliphatic hydrocarbons and esters as alternative solvents to the classical aromatic hydrocarbon mixture. The traditional and alternative Laropal® A81 varnish formulations based on Shellsol® A and IBIB, respectively, were assessed in their short- and long-term behaviours. A multi-technique approach including spectroscopic methods was applied to evaluate the optical, colorimetric, and stability properties of the formulations on inert (laboratory glass slides) and paint (mock-ups) supports before and after natural and accelerated light ageing. Their comparison served to assess the suitability of the alternative formulation. Extraction tests on naturally aged unvarnished paint films were also carried out to investigate the effect of both traditional and alternative formulation in terms of leaching of the paint film by the Laropal® A81 varnishes.

## 2. Materials and Methods

### 2.1. Materials

#### 2.1.1. Solvents

Shellsol® A (Kremer Pigmente, Aichstetten/Allgaeu, Germany) is a mixture of aromatic hydrocarbons and was used to prepare traditional Laropal® A81 formulation. It consists mainly of aromatic hydrocarbons (content > 97%) containing C9, low amount of benzene, cumene and the three isomeric forms of xylene (di-methyl-benzene ortho-, meta- and para).

Shellsol® D40 (Kremer Pigmente, Kremer Pigmente, Aichstetten/Allgaeu, Germany) is classified as an aliphatic mineral spirit and it is a mixture of C9-C11 paraffinic and naphthenic hydrocarbons, n-alkanes, isoalkanes, and cycloalkanes, with an aromatic content < 2% (D means dearomatised and 40 indicates the flash point). It has been used in both formulations as a cosolvent. It is unable to dissolve the resin by itself.

IBIB (≥97%) (2-methylpropyl 2-methylpropionate or isobutyl iso-butyrate) was purchased from Sigma-Aldrich and was used as the solvent in the preparation of the alternative aromatic-free formulation.

All solvents were characterised by $^1$H-NMR and IR spectroscopy to check their composition and to monitor their presence after the formation of the varnish film.

#### 2.1.2. Laropal® A81

The LMW Laropal® A81 synthetic resin (Kremer Pigmente, Kremer Pigmente, Aichstetten/Allgaeu, Germany) is a urea-aldehyde resin, marketed in solid grain form and described as highly lightfast by the manufacturer [16,28,29].

### 2.1.3. Tinuvin® 292

Tinuvin® 292 (bis(1, 2, 6, 6-pentamethyl-4-piperidyl) sebacate and methyl(1, 2, 6, 6-pentamethyl-4-piperidyl) sebacate) was purchased from Kremer Pigmente (Kremer Pigmente, Aichstetten/Allgaeu, Germany) and used as a formulation additive (at 2% of the dry weight of the resin) to add more stability towards the ultraviolet (UV) and visible (Vis) radiations of the varnish films [1,6,7,30,31].

### 2.1.4. Pigments

Cerulean blue (cobalt stannate, code 962, Zecchi, Florence, Italy), zinc white (ZnO, code 785, Zecchi, Florence, Italy), ivory black (calcium phosphate containing carbon, code 1200, Kremer Pigmente, Kremer Pigmente, Aichstetten/Allgaeu, Germany), and titanium rutile white (TiO$_2$, code 224227, Sigma-Aldrich, St. Louis, MI, USA) were employed as pigments in the preparation of the paint samples. The cerulean blue and the two white pigments were chosen to be able to clearly highlight any possible yellowing phenomena of the paint, while on the black backgrounds it is possible to better observe its brightness and any changes in the reflection of light by the paint surface. All pigments were used for the preparation of the varnished samples, while for the extraction tests only the films available with adequate natural ageing were used (ivory black with egg yolk and titanium rutile white for linseed oil).

### 2.1.5. Binders

Linseed oil for paint supplied by Zecchi, Florence, Italy was used. The egg yolk was recovered from fresh eggs and used as reported in traditional artistic recipes.

### 2.2. Sample Preparation

### 2.2.1. Varnishes

Two different formulations based on Laropal® A81 were investigated. Initially, 1:4 g/mL resin:solvent concentrations of traditional (A) and alternative (B) formulations were prepared at room temperature. Laropal® A81 grains were encased in cotton gauze and dipped into the solvent mixture. It took approximately 24 h for the resin to dissolve completely, after which Tinuvin® 292 was added. The films produced by these formulations simulated a typical application found in real cases. However, the characterisation of such a thin layer of varnish through the spectroscopic techniques selected in this study was unsuccessful. For this reason, higher concentration formulations (resin: solvent 1:2) were considered and implemented. With these second formulations, a thick layer of varnish was applied to the mock-ups and laboratory glass slides. In this way, they were eventually found to be suitable for this study, despite not being comparable to those actually applied on paintings. The composition of formulations A and B with 1:2 concentrations are presented in Table 1. The difference in the new formulation is the replacement of Shellsol® A by IBIB.

**Table 1.** Composition of varnish formulations A (traditional) and B (alternative).

| Ingredient | Formulation A | Formulation B |
|---|---|---|
| Laropal® A81 | 40 g | 40 g |
| Shellsol® D40 | 52 mL | 52 mL |
| Shellsol® A | 28 mL | - |
| IBIB | - | 28 mL |
| Tinuvin® 292 | 0.8 g | 0.8 g |

### 2.2.2. Varnished Samples (Painted Mock-ups and Transparent Glass Slides)

The varnish formulations were applied by brush to both laboratory glass slides (considered an inert support) and mock-ups. The latter were prepared by brushing a layer with one pigment at a time using egg yolk as a binder on a canvas support board (Michelangelo®)

(Figure 1; Table 2). The canvas support board substrate consisted of cardboard covered by a fine canvas frame of pure cotton (100%) with a dolomite preparation and an acrylic binder. Varnish layers without pigments were applied on glass slides to investigate the behaviour of formulations A and B with and without pigments (Table 2).

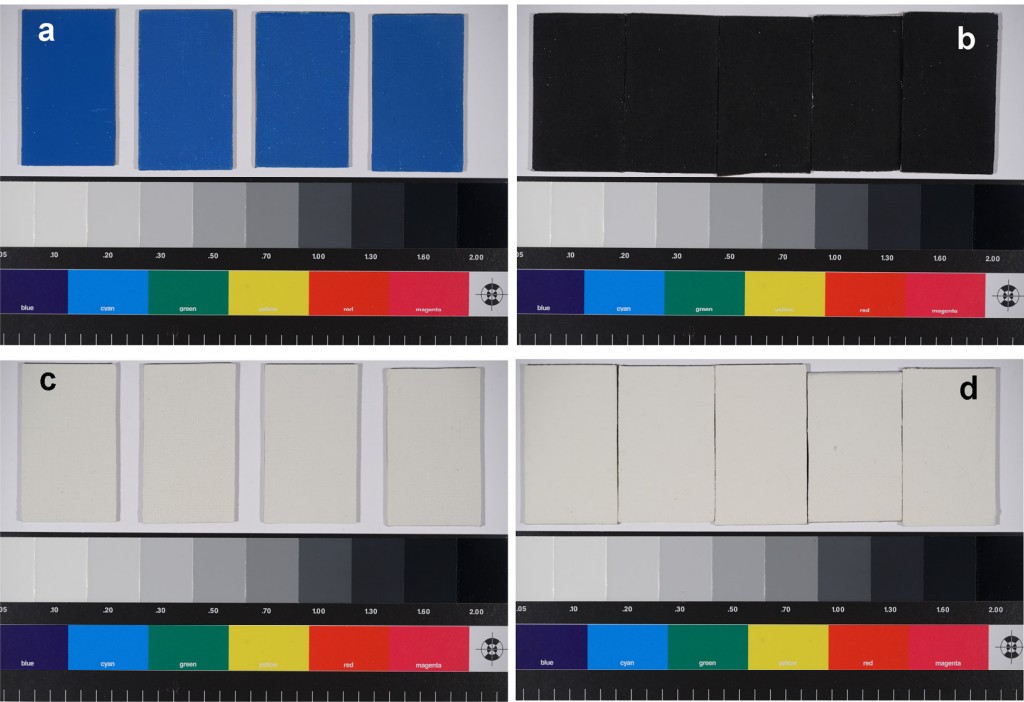

**Figure 1.** Pre-varnished mock-ups (5 cm × 3 cm): (**a**) cerulean blue; (**b**) ivory black; (**c**) zinc white; and (**d**) rutile. The samples were labelled with a letter (C-cerulean blue, A-ivory black, Z-zinc white, T-rutile) and a sequential number to identify them.

**Table 2.** Number of samples varnished.

| Samples with Each Formulation (A or B) |
| --- |
| 1 mock-up with cerulean blue |
| 1 mock-up with zinc white |
| 2 mock-ups with ivory black |
| 2 mock-ups with rutile |
| 3 glass slides |

### 2.2.3. Painted Mock-ups for Extraction Tests

Paint films were prepared as mock-ups using egg yolk or linseed oil as binders and were stored in the laboratory at room temperature. Fragments of these films were used for extraction tests after natural ageing (18 months for egg yolk and 12 years for linseed oil).

### 2.3. Analyses

The investigations of the samples were carried out by means of UV-Vis and near infrared (NIR) spectroscopy, Fourier transform infrared spectroscopy (FT-IR), [1]H-NMR spectroscopy, and colorimetry. This study was supplemented by photographic documentation of luminescence induced by UV radiation (UVL), diffuse RGB (Vis), and digital microscopy. To evaluate the formulations behaviour, all the samples were analysed at different times: (a) as soon as the varnish was dry, (b) after natural ageing, (c) after artificial ageing.

### 2.3.1. UVL-Vis Photographic Documentation

UVL-Vis photographic documentation was conducted using a Nikon D600 (Nikon, Tokyo, Japan) digital camera with a Nikon FX AF-D 50 mm lens and two modified LED lamps, model CTS ART LUX 100LW9 (C.T.S. S.r.l., Altavilla Vicentina (VI), Italy). During UVL acquisitions, a KV418 filter by Schott (SCHOTT, Jena, Germany) was mounted in front of the lens to cut off reflected UV radiation. To eliminate parasitic visible radiation, the lamps were modified at IFAC-CNR and fitted with the UG11 filter that cuts off VIS radiation and transmits UV radiation in the 300–390 nm range. The imaging geometry used for both configurations was $2 \times 45°/0°$ (two lamps positioned, on the left and on the right, at 45° with respect to the camera line of acquisition).

### 2.3.2. Dino-Lite Premier Digital Microscope

A Dino-Lite Premier digital microscope (AnMo Electronics Corporation, New Taipei City, Taiwan), using the DinoCapture 2.0 program, was utilised to collect images with magnifications at $20\times$, $50\times$, and $230\times$.

### 2.3.3. UV-Vis-NIR Spectroscopy

A Perkin-Elmer (Waltham, MA, USA) model Lambda 1050 UV-Vis-NIR spectrophotometer, equipped with two interchangeable modules, was used to acquire reflectance and transmittance spectra to characterise the paint samples (canvas cardboard, pigments, and varnish films) and the glass slides with the varnish layers, respectively. The reflectance module is equipped with a 60 mm diameter integrating sphere internally coated with a highly reflective surface (Spectralon®). The measurement configuration was $0°/d$ and only diffuse radiation was collected by the detectors integrated in the sphere. Reflectance spectra were acquired in the 200–2500 nm range with a sampling step of 1 nm and scanning speed of 196.14 nm/min. For transmittance acquisitions, glass sample spectra were recorded in the 320–2500 nm range, with an analysis step of 1 nm and scan speed of 210.97 nm/min. The spectra were acquired using the PerkinElmer UV WinLab™ 6.0.3.0730 software.

### 2.3.4. Fourier Transform Infrared Spectroscopy (FT-IR)

The FT-IR ALPHA Bruker Optics Inc. (Billerica, MA, USA) portable spectrometer was employed in this study. It was equipped with a transmittance module and an External Reflection module for the acquisition of total reflectance (TR) spectra. The reflectance mode was used to characterise the supports and the pigments and, subsequently, the varnish of the samples (both mock-ups and slides) during the ageing processes (unaged, naturally aged, and artificially aged). The transmittance mode was mainly used to characterise the purchased materials (resin, solvents, Tinuvin® 292, and canvas cardboard). All the spectra were acquired using OPUS 7.0.122 software from Bruker Optics Inc., with a resolution of $4\ cm^{-1}$, in the 7500–375 $cm^{-1}$ range with 256 scans for TR FT-IR measurements, and in the 4000–375 $cm^{-1}$ range with 64 scans for T FT-IR measurements. For the transmittance spectra, samples were dispersed in KBr (Aldrich, FT-IR grade), obtaining 13-mm pellets. Data processing was performed using EZ OMNIC 7.3 software. The Kramers–Kronig algorithm was applied to reflectance spectra in the 4000–375 $cm^{-1}$ range, when necessary, for the reconstruction of pseudo-absorbance spectra.

### 2.3.5. $^1$H-NMR Spectroscopy

A Varian Mercury Plus 400 MHz model NMR spectrometer or Varian INOVA 400 MHZ model NMR spectrometer (Varian Inc., Palo Alto, CA, USA) was used to characterise Laropal® A81, Shellsol® D40, Shellsol® A, IBIB, Tinuvin® 292, and the films obtained from the formulations A and B prepared by the different mixing of these components. This made it possible to check the stability of the formulations and to evaluate the presence of traces of solvents and additive after film formation. Furthermore, this technique was used to evaluate the possible extraction of binder components (egg or oil) from the mock-ups by the solvent mixtures A and B. Both instruments were equipped with a field gradient module,

and a temperature control and variation module. Spectrometers were used at a frequency of 399.921 MHz to record $^1$H-NMR spectra, and the samples were dissolved in deuterated chloroform (CDCl$_3$, Sigma Aldrich, purity 99.8%). Spectra were acquired using VNMRJ 1.1d software and were processed with Mestre-C 4.9.9.6 software (1996, Mestrelab Research S.L., Santiago de Compostela, Spain). They were reported in ppm and tetramethylsilane (TMS) was used as the external standard.

### 2.3.6. Colorimetry

Colorimetric measurements were carried out with the Konica-Minolta CM-700d spectrocolorimeter (Konica-Minolta Inc., Tokyo, Japan) to characterise the mock-ups without varnish (T$_i$), with varnish (T$_0$), after natural ageing (T$_1$) and artificial ageing (T$_2$). A transparent plastic mask was used to reposition the instrument over the same area when repeating the measurements (i.e., before and after ageing) and to avoid placing the spectrocolorimeter probe head directly in contact with the surface of the samples. Each measurement provided by the instrument was the result of the average of three acquisitions. Spectra were acquired on 8 mm diameter areas (MAV) with both specular components included (SCI) and excluded (SCE) and a measurement geometry of di:8° and de:8°. Colorimetric values were calculated for CIE D$_{65}$ standard illuminant and 1964 Supplementary Standard Observer (10°). Three measurements were recorded for each sample and their average was calculated. Data were processed using Spectra Magic NX software and Excel; in addition, the formula of $\Delta E_{00}$ (CIE 2000) was used to calculate colorimetric differences.

### 2.3.7. Extraction Tests

Extraction tests were performed on the unvarnished paint films to verify if the solvent mixtures used in both formulations (2.1.1) would leach them. Fragments of paint films prepared as mock-ups, stored in the laboratory, and naturally aged based on egg or oil as a binder were selected regardless of the type of pigment present. Fragments of linseed oil paint pigmented with rutile and egg tempera paint (yolk only) pigmented with ivory black and without pigment were soaked in two different solvent mixtures: (A) 65% Shellsol$^®$ D40 and 35% Shellsol$^®$ A, and (B) 65% Shellsol$^®$ D40 and 35% IBIB, imitating the solvent content of the traditional and alternative varnish formulations, respectively. The soaking tests were performed in accordance with tests reported in the literature [32]. Fragments of paint films (20–24 mg), previously prepared as mock-ups, were immersed in 2 mL solvent for a short (10 min) or long (24 h) time. The fractions extracted in the solvent mixtures were separated from the solid by filtration, evaporated to dryness, weighed, and analysed by $^1$H-NMR spectroscopy.

### 2.3.8. Natural Aging

Both sets of samples were aged under daylight at room temperature for one year. During this period, the samples remained inside containers with a transparent plastic lid to protect them from dust.

The original varnish formulation batches used for the preparation of the inert and paint samples were stored in the warehouse for about a year after application. Aliquots were taken after 3, 9, and 12 months, applied on a glass slide and, after drying, analysed to detect possible alterations.

### 2.3.9. Artificial Ageing

After natural ageing, one sample per type for a total of 10 (2 slides and 8 mock-ups) was subjected to accelerated ageing.

The CO.FO.ME.GRA model SOLARBOX 3000e climatic chamber (CO.FO.ME.GRA S.r.l., Milan, Italy) was used to perform an accelerated ageing test. The chosen environmental parameters had to simulate the conditions of irradiation, temperature and humidity typically found in a museum environment as closely as possible. The climatic chamber was equipped with a 2500 W xenon lamp and UV filters. During accelerated aging, an

indoor filter ($\lambda \geq 310$ nm) was used to simulate daylight coming through windows. The total irradiance was equal to 550 W/m$^2$ (corresponding to 105'000 lx and 54 W/m$^2$ of UV) with a temperature between 25 and 30 °C. The samples were exposed for 360 h (15 days for 24 h) receiving a total dose of 717 MJ/m$^2$. This treatment corresponds to approximately 87 years of ageing under museum conditions, considering only the visible component.

## 3. Results and Discussion

### 3.1. Materials Characterisation

Prior to the preparation of the formulations and their application, all ingredients and pigments were analysed to check their composition and purity as claimed by the manufacturer by FT-IR, $^1$H-NMR, or UV-Vis-NIR spectroscopies as required by each specific component. The spectra recorded on commercial products agreed with what was reported in the technical data sheets and in the literature, as reported in Table 3.

**Table 3.** Absorption bands and chemical shifts obtained for the varnish ingredients and pigments.

| Materials | IR Spectroscopy (cm$^{-1}$) | $^1$H-NMR Spectroscopy (ppm) | UV-Vis-NIR Spectroscopy (nm) |
|---|---|---|---|
| Laropal® A81 | 2963, 2873, 2720, 1732, 1650, 1487, 1443, 1368, 1309, 1263, 1216, 1154, 1086, 1006, 845, 756 | 0.98 (m), 1.20 (m), 3.1-3.5 (m), 3.67 (m), 4.26 (m), 4,43 (m), 4.99 (m), 5.15 (m), 9.53 (s) | 1190 (sh), 1700 (sh) |
| Tinuvin® 292 | 3000–2800, 1735, 1400–1300 | 1.07 (m), 1.16 (m), 1.24 (t, $J_{HH}$ = 7.4 Hz), 1.46 (m), 1.60 (m), 1.81 (m), 2.23 (m), 3.65 (m), 5.04 (m) | - |
| Shellsol® D40 | 3000–2800, 1460–1378 | 0.86 (m), 1.25 (m), 1.55 (m) | - |
| Shellsol® A | 3020, 2730, 1600–1500, 1500–1400, under 800 | 0.94 (m), 1.22 (m), 1.55 (s), 1.65 (m), 2.22 (m), 2.28 (m), 2.34 (m), 2.63 (m), 2.92 (m), 6.81 (m), 6.91 (m), 6.96 (m), 7.01 (m), 7.10 (m), 7.18 (m), 7.25 (m) | - |
| IBIB | 2950–2850, 1480–1340, 1195–1156, 1076–999 | 0.92 (d, $J_{HH}$ = 6.7 Hz), 1.16 (d, $J_{HH}$ = 7.0 Hz), 1.92 (m), 2.55 (m), 3.84 (d, $J_{HH}$ = 6.6 Hz) | - |
| cerulean blue | 650, 600, 455 | - | 560, 600, 650, 1280, 1500, 1800 |
| ivory black | 2020, 1050, 605, 560 | - | |
| zinc white | 500–400 | - | 384 (first derivative position) |
| rutile | 800–400 | - | 400 (first derivative position) |

In addition, preliminary extraction tests were performed to verify the possibility of extraction of paint components by the solvent mixtures used for the two formulations. The tests were performed by soaking, for short (10 min) and long (24 h) time fragments of egg yolk film, egg tempera paint (yolk only) pigmented with ivory black, and linseed oil paint pigmented with rutile in the two different solvent mixtures, A and B, used to obtain the traditional and alternative formulations, respectively. For these preliminary extraction tests, available films with adequate natural ageing samples (containing ivory black with egg yolk and titanium rutile white for linseed oil) were used.

The extracted fractions, separated from the solid by filtration and evaporated to dryness, were weighed and analysed by means of $^1$H-NMR spectroscopy to investigate possible leaching phenomena caused by A and B solvent mixtures. The quantitative results were comparable for both solvent mixtures but changed for each type of sample extracted (egg film as is, egg with pigment, and oil film with pigment) (Table 4). The extractions carried out on the egg yolk layers with ivory black and without pigment showed the signals

attributable to egg oil (Figure 2), which is soluble in chloroform, in a higher amount in the case of the yolk-only. In fact, the signals attributable to the saturated and oleic chains together with those attributable to the glyceric group, confirm the presence of soluble triglycerides in agreement with the non-drying properties of egg oils. Small variations in the relative intensity of the allylic signals found between samples of different compositions are attributable to the different effect of the pigment on the reactivity of these positions and not to the type of solvent used. On the other hand, the data obtained with the rutile pigment mixture with linseed oil could not be evaluated because the extractions were unable to highlight soluble components in accordance with the cross-linking of the binder and the reduced concentration of soluble components of the siccative oil. It is also necessary to consider that, for this pigment, the amount of binder used is extremely small and this factor also contributes to reducing the amount of soluble fraction (<1% as percentage relative to the dry weight (mg) of the extracted sample (mg)). In any case, no differences in behaviour between the two types of solvent mixtures, A and B, were noticed. The increase in the extraction time determines a slight increase in the quantities of fractions extracted in solution for the samples with egg binder while the values observed after 24 h for the samples with oil and rutile are substantially comparable and in agreement with the reduced presence of soluble components (<1%). The data obtained are in agreement with the composition and solubility of the samples such as the type of binder (egg or oil) and the amount of binder present in them with respect to the pigment (egg or egg with pigment).

**Table 4.** Extraction tests.

| Sample [1] | Weight (mg) | Solvent Mixture | Time | Extracted Fraction (%) [2] |
|---|---|---|---|---|
| egg 1 | 20.3 | A | 10 min | 40 |
| egg 2 | 20.9 | B | 10 min | 39 |
| egg 3 | 20.8 | A | 24 h | 49 |
| egg 4 | 21.3 | B | 24 h | 49 |
| black 1 | 22.1 | A | 10 min | 17 |
| black 2 | 23.6 | B | 10 min | 14 |
| black 3 | 21.7 | A | 24 h | 27 |
| black 4 | 20.7 | B | 24 h | 23 |
| white 1 | 20.4 | A | 10 min | <1 |
| white 2 | 21.3 | B | 10 min | <1 |
| white 3 | 20.0 | A | 24 h | <1 |
| white 4 | 22.2 | B | 24 h | <1 |

[1] egg = egg yolk without pigment, black = egg yolk with ivory black, white = oil with rutile; [2] percentage relative to the dry weight (mg) of the extracted sample (mg).

Since both solvent mixtures showed identical leaching behaviour, the next step was dedicated to studying the varnish films obtained with formulations A and B by FT-IR and $^1$H-NMR. The spectra recorded on the varnish films obtained from the two formulations showed the characteristic signals of the Laropal® A81 resin. Spectra recorded after one year showed no variations in the signals present and in their relative intensity, highlighting the stability of both formulations in storage at room temperature.

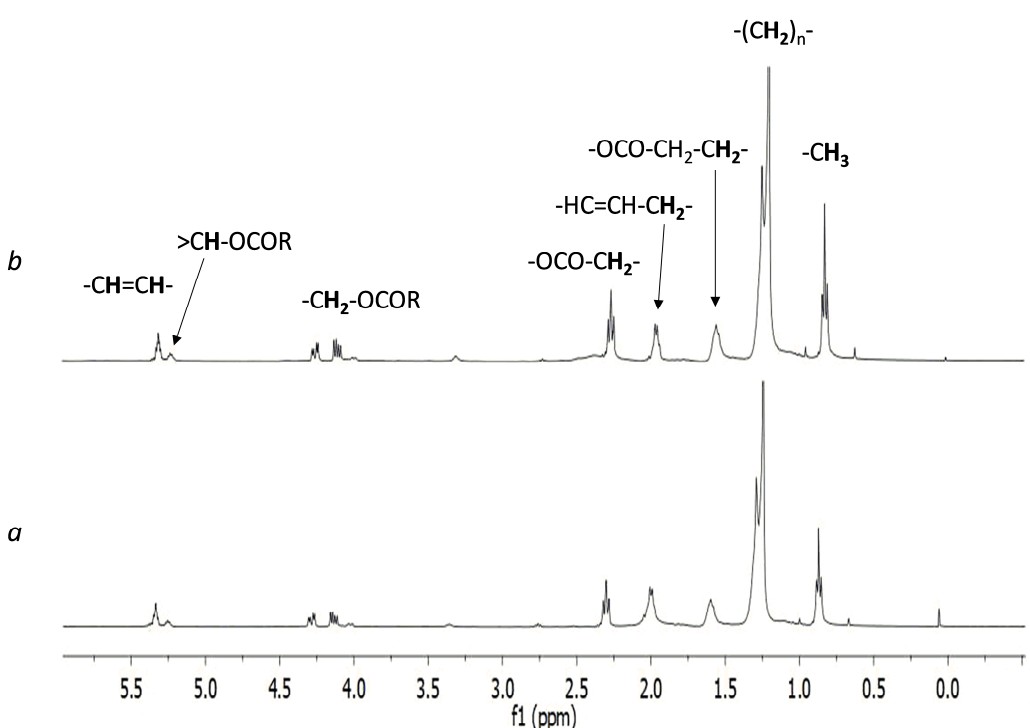

**Figure 2.** Extraction carried out on egg yolk with ivory black paint layers: (*a*) with solvent mixture of formulation A for 24 h; (*b*) with solvent mixture of formulation B for 24 h.

### 3.2. Study of the Samples with Varnish

The varnish formulations were applied by brush. Consequently, the varnish films were not flat but wavy and without a uniform thickness, as evinced by UVL (Figure 3).

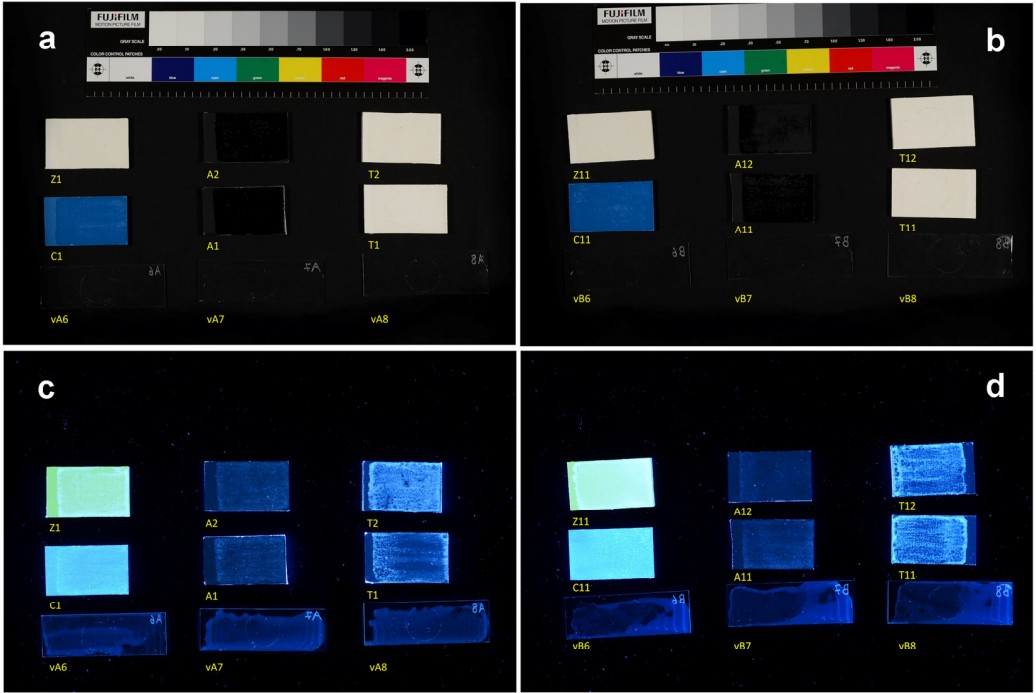

**Figure 3.** RGB and UVL images of varnished mock-ups and glass slides with A formulation (**a**,**c**), with B formulation (**b**,**d**), respectively.

The colorimetric data shown in Table 5 are the average of three measurements, and the maximum error in all measurements for each colorimetric value is $\leq 0.04$. L*, a*, and b* calculated before and after varnish application ($T_0$–$T_i$) on the slides and mock-ups of titanium white, showed non-significant differences. On the other hand, the two mock-ups of zinc white were characterised by a significant change in the b* coordinate; those of cerulean blue were also characterised by significant changes in the L* value, mainly for measurements in the SCE configuration, and partly in the a* coordinate for SCI measurements. Finally, it is interesting to note the result obtained for the A2 ivory black mock-up, where the colour difference $T_0$–$T_i$ shows a substantial difference between SCI and SCE measurements. The variation in Chroma ($C^* = [(a^*)^2 + (b^*)^2]^{\frac{1}{2}}$) after the application of the varnish cannot be very likely attributed to the varnish itself because a similar change in $C^*$ value would also have been detected on the slides ($T_i$ = laboratory glass slides without varnish; $T_0$ = varnished laboratory glass slides).

**Table 5.** Colorimetric variation of the samples. $T_i$: unvarnished samples; $T_0$: varnished samples; $T_1$: after natural aging; $T_2$: after artificial aging. C: cerulean blue; A: ivory black; Z: zinc white; T: titanium white; v: glass slide.

| Sample | | $T_0$–$T_i$ | | | | $T_1$–$T_0$ | | | | $T_2$–$T_1$ | | | | Form. |
|---|---|---|---|---|---|---|---|---|---|---|---|---|---|---|
| | | $\Delta L^*$ | $\Delta a^*$ | $\Delta b^*$ | $\Delta E_{00}$ | $\Delta L^*$ | $\Delta a^*$ | $\Delta b^*$ | $\Delta E_{00}$ | $\Delta L^*$ | $\Delta a^*$ | $\Delta b^*$ | $\Delta E_{00}$ | |
| C1 | SCI | −0.44 | 1.56 | −3.43 | 1.70 | −0.47 | 0.23 | 0.49 | 0.48 | 0.52 | 1.65 | −2.50 | 1.37 | A |
| | SCE | −3.45 | 0.12 | −5.69 | **3.72** | −1.35 | −0.21 | −0.19 | 1.17 | 3.72 | 3.28 | −0.09 | **3.84** | |
| C11 | SCI | −0.92 | 1.97 | −3.56 | 2.02 | 0.14 | 0.10 | 0.22 | 0.16 | 1.13 | 0.68 | −1.58 | 1.23 | B |
| | SCE | −3.20 | 0.94 | −5.30 | **3.52** | −0.44 | −0.18 | −0.20 | 0.40 | 3.64 | 1.86 | 0.31 | **3.45** | |
| A2 | SCI | 0.22 | −0.05 | −0.22 | 0.27 | 0.07 | −0.04 | −0.02 | 0.08 | 1.24 | 0.05 | −0.07 | 0.89 | A |
| | SCE | −12.05 | −0.10 | 0.24 | **8.01** | 1.83 | −0.02 | −0.19 | 1.17 | 10.16 | 0.08 | −0.30 | **6.82** | |
| A12 | SCI | 0.91 | −0.04 | −0.14 | 0.67 | −0.18 | −0.02 | −0.02 | 0.13 | −4.95 | 0.02 | 0.14 | **3.45** | B |
| | SCE | −0.25 | −0.07 | −0.09 | 0.23 | 0.49 | 0.00 | −0.05 | 0.35 | −4.32 | −0.01 | 0.11 | **3.00** | |
| Z1 | SCI | 0.13 | −1.02 | −6.65 | **4.75** | −0.21 | 0.06 | −0.56 | 0.45 | 0.46 | 0.03 | −2.67 | 2.17 | A |
| | SCE | −0.94 | −1.04 | −6.40 | **4.61** | −0.05 | 0.06 | −0.62 | 0.47 | 0.37 | 0.03 | −2.70 | 2.18 | |
| Z11 | SCI | 0.47 | −1.00 | −6.48 | **4.62** | −0.11 | −0.02 | −0.50 | 0.39 | 0.27 | 0.19 | −2.83 | 2.27 | B |
| | SCE | −0.71 | −1.02 | −6.19 | **4.43** | −0.13 | −0.01 | −0.53 | 0.40 | 0.19 | 0.19 | −2.88 | 2.29 | |
| T2 | SCI | −0.29 | 0.06 | −0.79 | 0.60 | −0.12 | 0.09 | −0.44 | 0.35 | 0.56 | −0.09 | −2.45 | 1.95 | A |
| | SCE | −1.18 | 0.03 | −0.58 | 0.82 | −0.09 | 0.10 | −0.45 | 0.36 | 0.40 | −0.11 | −2.47 | 1.94 | |
| T12 | SCI | −0.21 | 0.04 | −1.00 | 0.74 | −0.16 | 0.11 | −0.56 | 0.45 | 0.41 | −0.16 | −1.92 | 1.55 | B |
| | SCE | −1.29 | 0.01 | −0.74 | 0.95 | 0.08 | 0.12 | −0.65 | 0.50 | 0.46 | −0.16 | −1.95 | 1.57 | |
| vA7 | SCI | −1.07 | −0.09 | 0.18 | 0.65 | −0.59 | 0.07 | 0.28 | 0.45 | −1.24 | 0.01 | 1.06 | 1.23 | A |
| | SCE | −0.71 | −0.11 | 0.07 | 0.46 | −0.62 | 0.07 | 0.29 | 0.47 | −1.09 | 0.00 | 1.05 | 1.18 | |
| vB7 | SCI | −0.99 | −0.06 | 0.15 | 0.60 | −0.67 | 0.04 | 0.27 | 0.47 | −1.31 | 0.00 | 1.07 | 1.26 | B |
| | SCE | −0.55 | −0.08 | 0.09 | 0.36 | −0.73 | 0.05 | 0.30 | 0.53 | −1.26 | 0.00 | 1.11 | 1.28 | |

There are a number of studies that have looked at appearance effects due to the application of varnish films on paint layers [21–25,33]. Among them, the study by Simonot and Elias concludes that firstly the varnish layer induces a levelling of the paint layer, which results in an increase in specular reflected light and gloss [34,35]. Added to this is a decrease in diffuse reflectance due to the weak absorption of the varnish layer. The result, the authors argue, is a darkening and desaturation of the final paint surface the lighter and more saturated the initial paint film (and the thicker the varnish layer).

Here, the considerations of Simonot and Elias are partly observed. Indeed, the colourimetric differences found for the zinc white and cerulean blue mock-ups can be explained by observing their reflectance spectra acquired after their preparation ($T_i$) and after the

application of the thick layer of varnish ($T_0$) (Figure 4a,b). From their spectra, it is clearly visible that the Ti spectra in the UV-blue region show different trends compared to those of the same samples acquired at $T_0$. These differences are largely due to the curing processes of the paint film and partly to the presence of the superficial varnish film, which was very thick. In the case of the varnished zinc white mock-ups, this fact resulted in a reduction of the yellow component manifested in both the SCI and SCE measurements. For cerulean blue mock-ups, instead, the effect of the varnish is more evident in the SCE configuration since, in addition to a marked variation in b* (it tends to increase the perception of blue), and there is a perceptible decrease in brightness L*. Differently from what is reported by Simonon and Elias, in this case C* tends to increase with the presence of the varnish film. The case of one of the two ivory black mock-ups (A2) deserves attention since a very different behaviour of L* was found between the two measurement configurations. In fact, once painted ($T_0$) the lightness value changes a lot ($\Delta$L* = 12.05) compared to before ($T_i$) but only the value obtained with the SCE configuration. This fact can be explained by looking at the spectra (Figure 4c) in which it is evident that the varnish has considerably reduced the reflectance (SCE) over the entire visible range (as reported by Simonot and Elias). This effect is more perceived by the measure in which the specular component (SCI) is present as the new paint film is extremely smooth with a strong gloss component that results in a more intense reflectance.

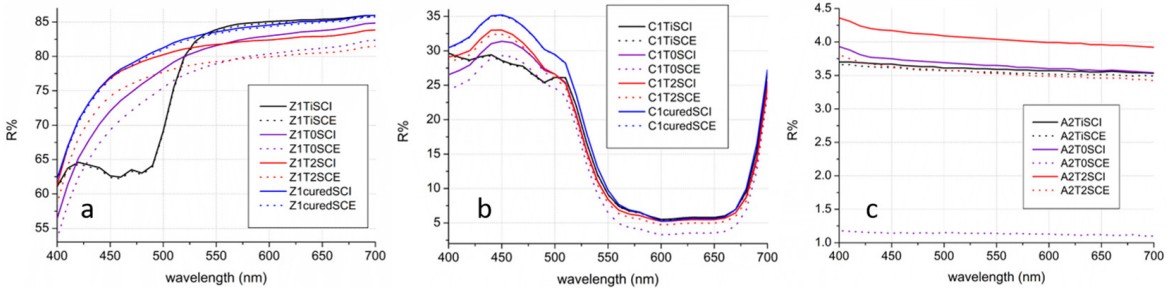

**Figure 4.** SCI and SCE reflectance spectra for zinc white (**a**), cerulean blue (**b**), and ivory black (**c**) mock-ups for Ti, T0, and T2. For zinc white and cerulean blue, the SCI and SCE spectra following Ti and before T0 where complete curing of the two paint films had been achieved are also shown.

TR FT-IR spectra of varnished glass slides and paint samples showed all the characteristic absorption bands of the Laropal® A81. No bands attributable to the underlying paint layer were detectable, probably because they are hidden by the varnish signals. As an example, the TR FT-IR spectra of the zinc white paint sample uncoated, with formulation A and with formulation B, are shown in Figure 5.

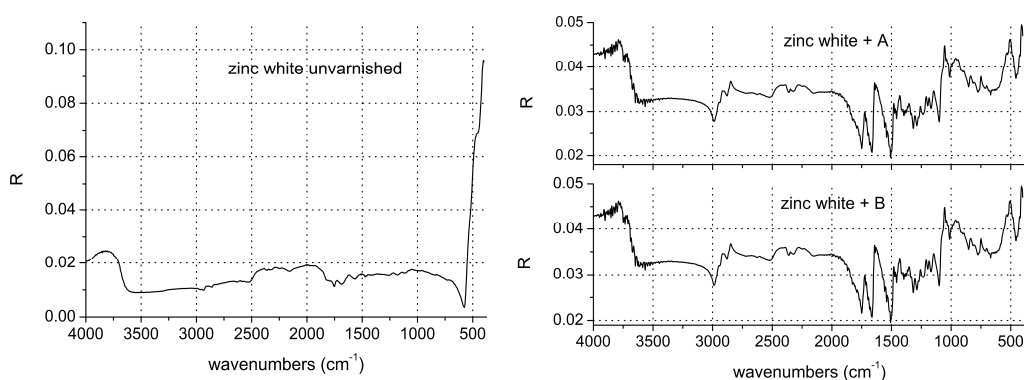

**Figure 5.** TR FT-IR spectrum of the zinc white mock-up uncoated (**left**), with formulation A (**up right**) and with formulation B (**down right**).

No significant differences in colorimetric data were found between the application of type A and type B formulations, and the same spectral trends were obtained from the two formulations.

### 3.3. Study of the Samples after Natural Ageing

The natural aging of the specimens did not lead to changes that were observable visually nor by the spectroscopic techniques. The recorded colorimetric data ($T_1$) showed irrelevant differences compare to $T_0$, and the total colorimetric variations ($\Delta E_{00}$) on all samples were considered imperceptible to the eye (Table 5). The TR FT-IR spectra, repeated after one year of natural ageing, were found to be completely superimposable with those recorded at $T_0$.

### 3.4. Study of the Samples after Artificial Ageing

The glass slides appeared unchanged under microscope magnification, and no spectral changes (FT-IR and UV-Vis-NIR) were observed. Although a slight yellowing was detected ($\Delta E_{00} \approx 1$; $\Delta b^* \approx 1$), it remained below the limit perceptible by the naked eye. This result confirmed the stability of Laropal® A81 urea-aldehyde resin applied alone on an inert substrate.

Some artificially aged mock-ups showed visually and analytically detectable differences (imaging, FT-IR, UV-Vis-NIR, and $^1$H-NMR). Furthermore, these were most likely influenced by the substrate (i.e., the pigments) rather than the different formulations (A or B). Interestingly, the varnish films remained transparent in all cases with no or little perceptible yellowing (see $T_2$ in Table 5), cracks, or lifting.

The zinc white mock-ups exhibited only a small reduction in the $b^*$ coordinate toward neutrality values, but the total colorimetric variation could also be considered almost imperceptible ($\Delta E_{00} \approx 2$). The TR FT-IR spectra were characterised by the fingerprint bands of Laropal® A81, and no bands attributable to structural changes in the oligomer or the presence of alteration products were detected.

A relatively good preservation of the varnish layer was also observed for the artificially aged titanium white paint samples. As with the zinc white mock-ups, a reduction in the yellow values ($b^*$) towards the neutrality was detected with a non-perceptible colour variation ($\Delta E_{00} < 2$). Laropal® A81 absorption bands were still clearly visible in the IR spectra, although, in some cases, they appeared to have a lower relative intensity because the characteristic bands of the titanium white pigment became more prominent (Figure 6). These data could be related to a thinning of the varnish film, as reported in other papers for Laropal® A81 on silicon wafers, quartz, and simple glass slides [36,37].

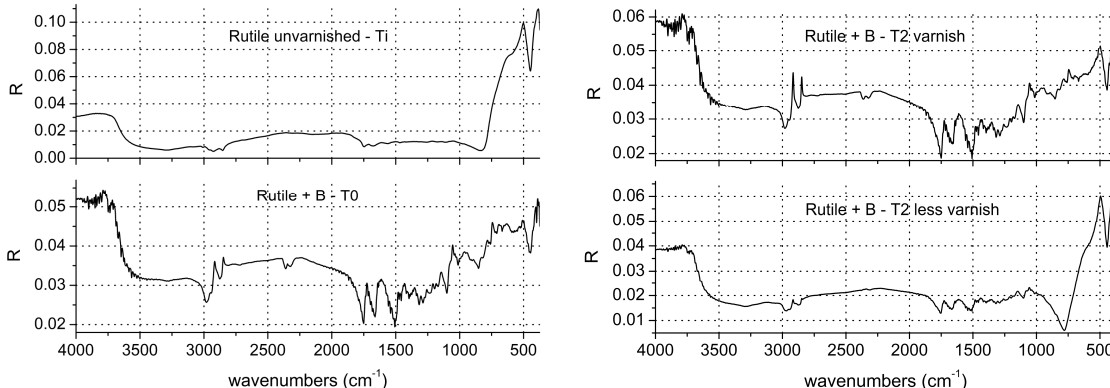

**Figure 6.** TR FT-IR spectra of a mock-up with titanium white pigment rutile unvarnished ($T_i$), with formulation B ($T_0$), and after artificial ageing ($T_2$) in two different locations where the varnish layer has a different thickness due to artificial ageing.

The cerulean blue and ivory black mock-ups showed the most noticeable changes following accelerated ageing, with significant colorimetric variations visible to the naked

eye. The colorimetric parameters indicate that the blue mock-ups became lighter and less greenish after ageing, in particular for the SCE measurements ($\Delta E \approx 3.5$). The two ivory black mock-ups, also in this case, show a different behaviour. In fact, the A12 mock-up features $\Delta E_{00} \approx 3.0$ for both SCI and SCE. On the other hand, the A2 mock-up confirms what was previously shown ($T_0$–$T_i$) given that the SCI component shows negligible $\Delta E_{00}$ values while the SCE component has $\Delta E_{00} \approx 7$, also in this case due to L*, bringing it back to have colorimetric values more similar to those of the unvarnished mock-up. This behaviour leads us to hypothesise that the artificial ageing caused a strong thinning of the varnish layer bringing the two mock-ups back to conditions similar to $T_i$.

The TR FT-IR and T FT-IR spectra collected from the blue and black mock-ups displayed no structural changes in the varnish or alteration products. However, they showed a less homogeneous distribution of both traditional and alternative varnishes after ageing, as the absorption bands of the resin show highly variable intensities depending on the areas analysed. This could be attributed to a decrease in its thickness in some areas caused by ageing in the climate chamber (Figure 7).

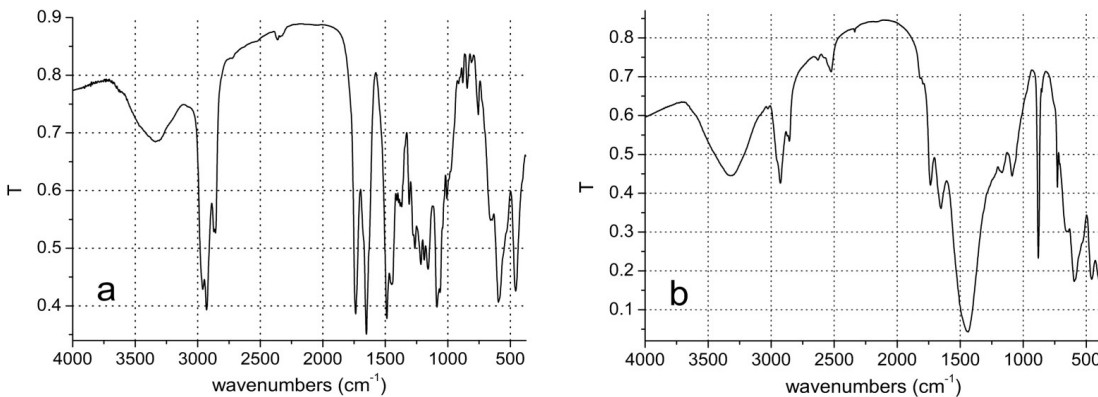

**Figure 7.** T FT-IR spectrum of a sample of the mock-ups with cerulean blue and formulation B after accelerated ageing: (**a**) taken from an area with varnish (bands due to the Laropal® A81 are still detectable); (**b**) taken from an area with small varnish residues (only very low signals attributable to Laropal® A81are detectable, while bands due to calcite of the substrate and pigment have become dominant).

To try to better understand the impact of the artificial ageing procedure (at $T_2$) on the ivory black mock-up, the one with varnish formulation A was analysed. To do so, the varnish residue from the surface was extracted using two cotton swabs, one soaked in ethyl acetate and one in acetone. The solutions obtained by counterwashing the two cotton swabs with the same solvents were taken to dryness, and the residue was analysed by ¹H-NMR (Figure 8a). Tinuvin® 292 (Figure 8b) is present in the varnish formulation in a small concentration as an additive, and before ageing (Figure 8c) the signals from Tinuvin® 292 (highlighted with *) were much less intense than those of the Laropal® A81 resin which were well recognised. After ageing, signals from Tinuvin® 292 were more intensely visible in spectrum *a* than those characteristic of the Laropal® A81 resin, which were weakly present. These results support the hypothesis of the varnish layer reduction, which cannot be attributed to a variation in the solubility of the resin after ageing. This is probably attributable to a reduction in the quantity of the varnish (particularly of the dried Laropal® resin) as a result of depolymerisation processes, with the resin becoming proportionally less concentrated than Tinuvin® 292. Spectrum *a* also shows other not easily attributable signals that could be assigned to components extracted from the paint film; however, it is difficult to hypothesise a connection with transformation products of Tinuvin® 292 or Laropal® A81. In accordance with this interpretation, some TR FT-IR spectra acquired on areas with significant resin reduction showed only the spectral features of the pigments, whereas no absorption bands associated with Laropal® A81 were detected. It is hypothesised that

depolymerisation of the varnish occurred, which caused its loss through the formation of volatile products, in agreement with evidence in the literature [36,37].

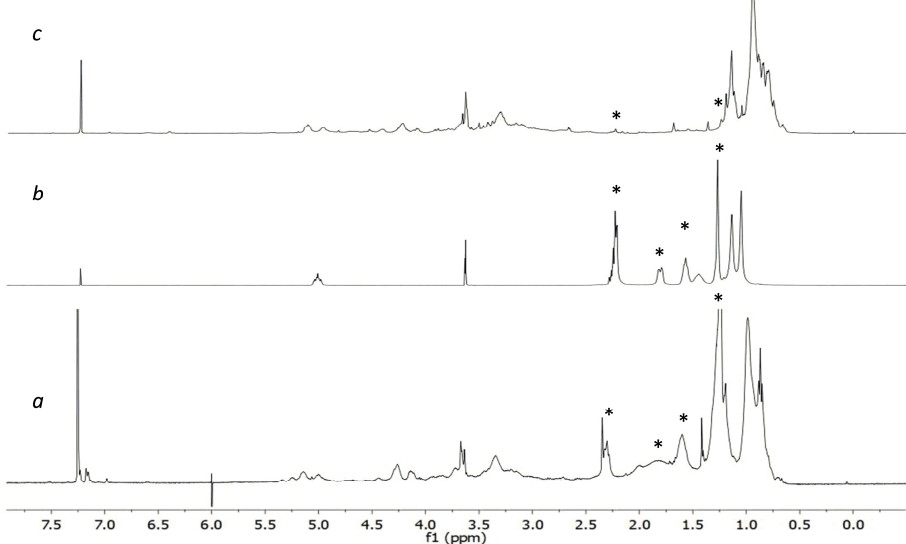

**Figure 8.** (*a*) $^1$H-NMR spectrum of the extract from mock-ups with ivory black and formulation A at T$_2$. (*b*) $^1$H-NMR spectrum of Tinuvin® 292. (*c*) $^1$H-NMR spectrum of film obtained from formulation A before ageing. * Tinuvin® 292 signals.

Observing the UV-Vis spectral trends of the mock-ups (Figure 9), one can infer that the film-forming materials applied to the glass slides transmitted the incident radiation without any absorption. As far as mock-ups are concerned, both white mock-ups absorbed the UV component while reflecting the visible one; those with cerulean blue absorbed the UV radiation and that in the 530–680 nm range; the black samples have absorbed all the incident radiation emitted by the SOLARBOX 3000e. It follows that the greater absorption of the incident radiation by the darker mock-ups could have induced a photochemical degradation and an increase in the temperature of the paint film surface that, in turn, would have increased the temperature of the varnish.

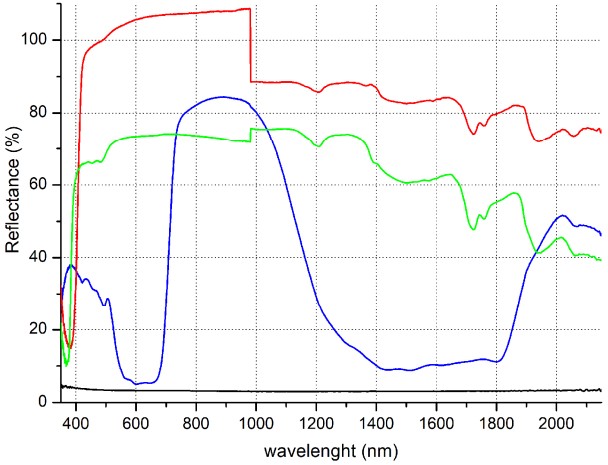

**Figure 9.** UV-Vis-NIR spectra of the painted layers without varnish. Blue line—Cerulean blue, black line—ivory black, green line—zinc white, and red line—rutile.

## 4. Conclusions

The most significant result obtained is that the study on the application properties of traditional and alternative Laropal® A81-based formulations, using chemical–physical techniques, showed no disadvantages between the application of type A and type B

formulation. The aromatic percentage (Shellsol® A) traditionally used with an aliphatic hydrocarbon (Shellsol® D40) to adequately solubilise Laropal® A81 can thus be replaced with the alternative solvent (IBIB) without any particular criticism of the painting films and with considerable advantages for the health of the operators and environmental protection.

Moreover, based on the data obtained, it can reasonably be stated that the stability of Laropal® A81 resin is strongly influenced by the substrate present while it is not influenced by the formulation used.

After accelerated ageing, the results lead to the hypothesis of a depolymerisation of the Laropal® A81 resin, with consequent loss of product due to the formation of volatile products. This phenomenon had already been detected by the studies of Farmakalidis et al. [36,37], in which a decrease in the thickness of Laropal® A81-based varnish films was found, confirming the decrease in the molecular weight of the resin already reported in previous studies [4,9,16,38]. Information obtained from the studies by Farmakalidis et al. suggests that the emission of volatile products could be related to the breaking of the N–(C=O) bond with the subsequent formation of formaldehyde and urea. This would indicate a partial depolymerisation of the material and would explain the decrease in film thickness [36,37]. In the current project, this phenomenon was further explored with a clear distinction between the different colour backgrounds. The results shown that the layers of paint remained virtually unchanged on the glass slides and on the white backgrounds, while a loss of resin was found on the blue and black backgrounds. Moreover, it is very important to note that only in cases of artificial ageing (corresponds to approximately 87 years of ageing under museum conditions) this behaviour was observed on part of the painted areas, while this phenomenon was not observed for natural ageing (12 months). In addition to the difficulty of relating the effects of artificial ageing to natural ageing and their temporal correspondence, it is important to remember that the use of artificial ageing procedures significantly shortens exposure times to specific environmental parameters, and that in many cases they tend to produce conditions that are not perfectly comparable to those found in natural ageing. As far as the purpose of this study is concerned, it is of absolute importance to note that the results obtained do not appear to be influenced by the application of either the type A or type B formulation. In cases where the varnish persists, the recorded spectra reveal the characteristic signals of Laropal® A81, which is therefore unaffected; no alteration or degradation products that could be harmful to paintings are detected in these areas. Furthermore, both A and B varnish films, even after accelerated ageing, remained transparent, with no perceptible yellowing and without cracking or lifting. In the future, it would be desirable to extend the studies on the stability of Laropal® A81 following natural ageing for a longer time than that applied in this project, since studies on the oligomer are mainly found in the literature only following its accelerated ageing [4,15,16,18,36,37].

Additionally, as for the extraction tests carried out to assess possible leaching attributable to solvent mixtures A and B, it is important to observe that there were no differences in behaviour between the two formulations. However, the results will deserve further investigation, increasing the number of samples (pigments and binders) studied. In fact, the data obtained, on fragments of mock-ups aged naturally for about a year and a half, did reveal an unsatisfactory phenomenon with respect to egg-yolk-only and ivory black samples with yolk. Soluble components (egg oils) were indeed extracted from these fragments, which may have an influence on the plasticity of the film. On the other hand, no protein-binding extractions were observed. Certainly, the phenomenon deserves attention and further study to assess whether the loss of non-cross-linked egg oils is a phenomenon that is nonetheless present in aged egg films. Studies conducted to assess leaching against egg tempera paints are underrepresented in the literature [39], while the phenomenon was mainly investigated on oil painting samples [10,32,40,41]. However, data from extraction tests conducted on paint samples prepared about 12 years ago using linseed oil and rutile do not show any appreciable results in terms of extraction, probably due to a very little amount of non-cross-linked materials present. With regard to the specific behaviour identi-

fied for the egg binder, it will be interesting in the future to compare the leaching of egg tempera samples containing different pigments and after different ageing conditions and times. Based on the preliminary results obtained, [1]H-NMR spectroscopy is able to provide in detail the effects on the reactivity of the allylic position, and based on this it will be interesting to confirm the effect of the different metals contained in the pigments.

Further developments and research on this subject could involve the study of Laropal® A81 in interaction with other types of substrates, both paint layers and support, as the substrate present has been found to be an influential factor in the preservation of Laropal® A81 layers. In future research, it might be of interest to measure the surface temperature of the samples during accelerated ageing (comparable with the one carried out in the present study) in order to detect possible heating of the samples, and it might be relevant to combine this type of ageing with artificial thermal ageing.

The results obtained in this research are specific for the Laropal® A81 or for resins with similar solubility. However, the proposed study procedure is also applicable to formulations with different resins and different solvent mixtures with the aim of replacing solvents that are hazardous to the health of operators and to the environment. As far as the formulations studied in this research are concerned, further studies will be able to expand the knowledge relating to leaching on different binders and on the basis of this behaviour evaluating how to improve it, if possible, in the choice of new formulations that are safer for the environment and operators.

**Author Contributions:** Conceptualization, I.P., A.S., V.C. and M.P. (Marcello Picollo); data curation, I.P., A.S., E.M.A., G.B. and M.P. (Marcello Picollo); formal analysis, I.P., A.S., E.M.A., G.B. and M.P. (Marcello Picollo); funding acquisition, A.S., M.P. (Marisa Pamplona) and M.P. (Marcello Picollo); investigation, I.P., A.S., E.M.A., G.B. and M.P. (Marcello Picollo); methodology, A.S., V.C. and M.P. (Marcello Picollo); project administration, A.S., V.C. and M.P. (Marcello Picollo); resources, A.S., M.P. (Marisa Pamplona), and M.P. (Marcello Picollo); supervision, A.S., V.C. and M.P. (Marcello Picollo); validation, I.P., A.S., E.M.A., M.P. (Marisa Pamplona), V.C. and M.P. (Marcello Picollo); writing—original draft, I.P., A.S., E.M.A., G.B. and M.P. (Marcello Picollo); writing—review and editing, I.P., A.S., E.M.A., M.P. (Marisa Pamplona), G.B. and M.P. (Marcello Picollo). All authors have read and agreed to the published version of the manuscript.

**Funding:** This research received no external funding.

**Institutional Review Board Statement:** Not applicable.

**Informed Consent Statement:** Not applicable.

**Data Availability Statement:** Not applicable.

**Acknowledgments:** We thank MIUR-Italy ("Progetto Dipartimenti di Eccellenza 2018–2022" for the funds allocated to the Department of Chemistry "Ugo Schiff").

**Conflicts of Interest:** The authors declare no conflict of interest.

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
