# Peer review of "A Formulation for a New Environmentally Friendly Varnish for Paintings"

_coatings, doi:10.3390/coatings13091566_

Round 1
Reviewer 1 Report
my summary of their work: The research investigates a new formulation for Laropala81, a urea-aldehyde resin used as a paint varnish, aiming to replace traditional aromatic solvents with less harmful alternatives. The study compared the conventional and alternative formulations by evaluating their optical, colorimetric, and stability properties before and after aging. The results indicate that the alternative formulation using aliphatic hydrocarbons and esters (iso-butyl isobutyrate, IBIB) showed no disadvantages and can safely replace the aromatic solvents, leading to a more sustainable conservation practice with benefits for the health of operators and the environment.
comments:
While the study found no differences in behavior between the two formulations (Type A and Type B) regarding leaching, there was a noteworthy phenomenon observed with egg yolk-only and ivory black samples with yolk. This phenomenon deserves further attention and study to understand its implications for aged egg films. Can you go into more detail about your hypotheses and potential ways this should be investigated in the future? Obviously, additional experiments are outside the scope of this work but your thoughts would be beneficial to the literature on this matter.
The literature referenced mainly contains studies on the oligomer following accelerated aging. More literature research on the properties and behavior of Laropal A81 after natural aging would be valuable to complement existing knowledge if it exists, this reviewer is not sure, and if it does not exist, a comment about that would be helpful in the manuscript.
The study focused on comparing traditional and alternative formulations of LaropalA81. A broader analysis could involve comparing the performance of Laropal A81 with other resins commonly used in varnishing and their respective environmental and health impacts. Please comment on the limitations due to this.
To strengthen the findings, future research could involve increasing the number of samples (pigments and binders) studied, providing more robust data for analysis. Please comment on the limitations of the conclusions due to this.
Reviewer 2 Report
Dear authors,
The presented work is highly intriguing from an experimental perspective and holds appeal for academics, conservation and heritage restoration professionals, and scientists at large.
However, significant methodological gaps are detected in the section concerning the colorimetric evaluation of the new varnish formulation proposed in your study. This factor is crucial for proper explanation and conclusion, as outlined in the objectives.
We would like to offer some suggestions.
Kind regards,
Line 45. Could you clarify the term "dispersion factor"? Is it recognized in scientific literature?
Line 339, Table 4. In the section "Residual weight of the extract (mg)," the values should be expressed in a dimensionless form, specifically as a percentage relative to the dry weight (mg) of the extracted sample.
Lines 350 to 361.
1. The explanation of colorimetric results is inconsistent and incongruent.
2. Generally, the presentation of a results table with values > 250 implies a detailed explanation, which is not provided in the research work.
3. Furthermore, it is stated that "insignificant" colorimetric variations are detected, even though there are DeltaE values exceeding "3," a threshold that multiple researchers consider as the limit of human visual detection.
4. The table itself is poorly organized as it highlights in bold the DeltaE (CIE 2000) values of T1-T0 and T2-T1, but not for T0-Ti. The table should highlight DeltaE values exceeding "3" in bold.
5. Are the values presented in the table the mean values? If so, the standard deviation of each recorded variation should be indicated.
In addition, the colorimetric incidents that occurred during the experimental phase are not adequately interpreted or explained.
6. The alteration observed after applying the varnish can't be readily ascribed to varnish tinting, as a comparable shift in chroma level would have been identified in the slides as well... This doesn't establish a cause-and-effect relationship.
7. However, it can be assumed that the varnish, once applied and dried, introduces a very weak contribution to the blue radiation due to the scattering of the paint film in such blue and white paint-film system, This should be scientifically elucidated, preferably with relevant bibliographic references.
8. Consequently, the colorimetric variation can probably be related to how Laropal® A81 saturates the colour of these paint samples, causing them to have a cooler hue. The colorimetric variation cannot be attributed to Laropal® A81 without specific experimentation conducted within the scope of this study.
9. It is not appropriate to provide a human sensory explanation (e.g., cooler hue) for a specific tonal variation. Instead, the specific variation should be quantified, preferably as the percentage change in the b* axis, indicating whether it is positive or negative.
It is mandatory to address the colorimetric study in a detailed and comprehensive manner, as it constitutes one of the main objectives of this research. This issue can be resolved by graphically representing the mean values on a chromaticity plane according to CIELab 1976, which will enable the detection of variations in hue, chroma, and saturation. Additionally, an auxiliary axis can be introduced on the x-y graph (axes a*+-, b*+-) corresponding to L*, which will facilitate the identification of potential luminosity variations. All these values represent the total color difference (E) across the various samples.
Conclusions, lines 478 to 541, should be enumerated and organized hierarchically.
Lines 564 to 626. The present research work should be supplemented with numerous bibliographic references, as there is a quantitative deficiency for enhanced scientific support.
Reviewer 3 Report
GENERAL COMMENTS
The work is focused on finding a less harmful alternative for replacing a common varnish formulation for paintings. The work is interesting, and the manuscript is fairly well written and organised.
The introduction is concise and cites relevant literature in the field.
However, there a few issues that should be addressed before publishing, as specified below.
SPECIFIC COMMENTS
Abstract: The authors write that “The results indicate that there were no disadvantages in the application of the alternative formulation” – I suggest writing a more positive sentence.
2.2.1. To improve clarity, I suggest to highlight that the main difference in the new formulation is the replacement of Shellsol A by IBIB.
2.1.4. I suggest stating the reasoning for selecting the listed pigments.
Table 2: Since both Samples with A and B formulation have an identical number of samples varnished, I suggest listing only one column or substituting the table with a paragraph..
2.2. It could also be interesting to include a blank sample without varnish and another with a traditional varnish (e.g., linseed oil) to check the effects of ageing.
2.3. In the subsections, I suggest beginning with the description of Colorimetry (same order as mentioned in the introductory paragraph of this section).
2.3.5. The use of oil (which type of oil?) was not mentioned previously (neither in the Materials or Sample preparation sections) and appears in this section “out of the blue” (there is a typo in the numbering of this section).
2.3.7. In this section, the authors mention that the extraction tests were performed on unvarnished paint films with linseed oil, but this information has not been mentioned previously, neither in the Materials nor Sample preparation sections. The previous comment on section 2.2. was avoidable.
2.3.9. For future work, it would be interesting to also include the results of accelerated ageing corresponding to 1 year of ageing to compare with the results of 1 year of natural ageing.
3. Results and discussion
Table 3: I think that this table should be presented as an appendix.
I do not understand the selection of the pigments for the extraction tests: why did the authors use different pigments for each binder?
Table 4: Can you please explain the relevant differences in the residual weight for the samples with oil? In particular, why the much longer extraction process resulted in a similar weight (formulation A) or much lower weight (B). Only 1 sample was analysed?
(there is a typo in the last line of the table: “whit” instead of “with”)
Lines 429-431: I do not understand how the authors infer that the distribution became less homogeneous from the FTIR data and how the ageing process can reduce the thickness of the varnish. Can you please explain better? I think that this aspect is rather attributable to the inhomogeneity of the layers resulting from the brush application technique (as the authors note in lines 350-351 while referring to Fig. 3).
The manuscript is fairly well written and organised.
Round 2
Reviewer 2 Report
Dear authors, I am grateful that your suggestions have been taken into account in order to improve scientific dissemination. My opinion is that the proposed article has been substantially improved for publication.
Kind regards.